# Triclosan to Improve the Antimicrobial Performance of Universal Adhesives

**DOI:** 10.3390/polym15020304

**Published:** 2023-01-06

**Authors:** Yubin Yang, Jingyu Ding, Xuanyan Zhu, Zilu Tian, Song Zhu

**Affiliations:** Department of Prosthetic Dentistry, Hospital of Stomatology, Jilin University, Changchun 130012, China

**Keywords:** triclosan, universal adhesive, antibacterial properties, bond strength

## Abstract

To solve the proble ms of composite restoration failure caused by secondary caries, this study reports a light curable antibacterial triclosan derivative (TCS-IH), which was synthesized and added to the existing commercial universal adhesive to achieve a long-term antibacterial effect The effect of mixing different mass percentages of TCS-IH on the bond strength of dentin was also investigated.TCS-IH was synthesized by solution polymerization and characterized by nuclear magnetic resonance hydrogen spectroscopy (^1^H NMR) and Fourier transform infrared (FTIR) spectroscopy. Two commercial universal adhesives, Single Bond Universal and All Bond Universal, were selected and used as the control group, and universal adhesives with different mass percentages (1 wt%, 3 wt%, 5 wt% and 7 wt%) of TCS-IH were used as the experimental group. The antibacterial properties were analysed by means of colony count experiments, biofilm formation detection, plotting of growth curves, biofilm metabolic activity detection, insoluble extracellular polysaccharide measurements and observations by confocal laser scanning microscopy and scanning electron microscopy (SEM). The effect of adhesives on biofilm formation, metabolism, extracellular matrix production, distribution of live and dead bacteria, and bacterial morphology of *Streptococcus mutans* (*S. mutans*) was analysed. The mechanical properties were evaluated by the degree of conversion and microtensile bonding strength under different conditions. Its biosafety was tested. We found that the addition of TCS-IH significantly improved the antibacterial performance of the universal adhesive, with the 5 wt% and 7 wt% groups showing the best antibacterial effect and effectively inhibiting the formation of biofilm. In addition, the adhesive strength test results showed that there was no statistical difference (*p* < 0.05) in the microtensile bond strength measured under various factors in all experimental groups except for the 7 wt% group in the self-etch bonding mode, and all of them had good biosafety. In summary, the 5 wt% group of antibacterial monomer TCS-IH was selected as the optimum addition to the universal adhesive to ensure the antimicrobial properties of the universal adhesive and the stability and durability of the adhesive interface. This study provides a reference for the clinical application of adhesives with antimicrobial activity to improve the stability and durability of adhesive restorations.

## 1. Introduction

Caries has become the most common oral disease, affecting the oral health of people of all ages to varying degrees [1]. Light curing composite resins are widely used in the repair of clinically decayed teeth due to their ideal physicochemical, mechanical, and biocompatible properties, as well as their good aesthetic properties [2]. Dental adhesive provides a bonding bridge between composite resins and dental hard tissues, mainly enamel and dentin. Universal adhesives are new types of adhesive agents developed in recent years. The purpose of universal adhesives is to concentrate multiple and two-bottle components into a single bottle, simplifying clinical procedures, reducing technical sensitivity, realizing one-step operation, and having the advantages of higher bonding strength, less dissolution and good colour matching. As a one-step self-etching system, the one-bottle system can be used for etch-and-rinse, self-etch or selective etching bonding modes.

In addition to the basic components of the adhesive, some universal adhesives contain acidic functional monomers such as 10-methacryloyloxydecyl dihydrogen phosphate (10-MDP), which exhibit good and relatively stable bonding strength both in the etch-and-rinse and self-etch bonding mode and immediately or after ageing. The phosphate group of 10-MDP has been reported [3] to have the ability to interact with hydroxyapatite and to play an important role in the long-term durability of the adhesive-dentin bonding interface. There are two main mechanisms of action of 10-MDP [4]: (1) 10-MDP forms a stable, water-insoluble calcium salt by electrostatic interaction with calcium ions in hydroxyapatite crystals; (2) the phosphate group in 10-MDP interacts chemically with the corresponding phosphate group in hydroxyapatite to form insoluble calcium salts. These calcium salts are deposited on the outer surface of the hydroxyapatite crystals, and the salt-containing coating is named the “nanolayer” [5]. Another functional monomer in the adhesive, polyalkenoic acid copolymer (PAC), interacts chemically with the calcium in hydroxyapatite [6]. Moreover, PAC improves the stability of the adhesive on wetted dentin matrix surfaces [7], which is important for the etch-and-rinse bonding mode.

However, secondary caries is still the main cause of adhesive restoration failure, with *S. mutans* as the main pathogen [8]. The aim of adhesive restorations is to form a favourable dentin-bonding interface, and the stability of the interface is a prerequisite for favourable bond strength, marginal sealing and clinical durability [9]. The bonding interface is continuously affected by water, protease and chewing pressure due to polymerization shrinkage and microleakage of the adhesive [10]. Moreover, due to the requirements of minimally invasive restorative dentistry, some residual bacteria are still located in the dentin matrix when removing carious lesions to preserve as much dentin tissue as possible [11]. Under the influence of these factors, the bond strength of the restoration decreases and promotes the attachment of plaque biofilm at the bonding interface, which in turn accelerates the failure of the adhesive restoration.

Therefore, it is urgent to prolong the life of adhesive restoration by inhibiting the growth of *S. mutans* and the formation of biofilms. Various antibacterial monomers have also been successively added to adhesives, such as quaternary ammonium compounds [12], metal ions such as silver and zinc [13], antibiotics [14], and antimicrobial peptides [15]. However, metal and its oxide particles will change the colour of the material and affect the aesthetics and resin transmittance [16]. Antibiotic resistance is a threat to public health. At the same time, quaternary ammonium compounds have been reported to be able to induce bacterial drug resistance [17].

Triclosan is already a widely used antimicrobial agent in oral hygiene care products, acting on both gram-negative and gram-positive bacteria [18]. In addition to its antibacterial properties, triclosan has a direct anti-inflammatory effect, inhibiting the arachnoid metabolizing enzymes that play a key role in the development of gingivitis [19]. Triclosan is also used in other fields, such as cosmetics, soaps, textiles and even in children’s products [20]. Toxicological studies have shown that triclosan and its metabolites are well tolerated by several species, including humans [21]. Triclosan is considered safe in reproduction studies; in addition, it has no carcinogenic, mutagenic or teratogenic activity [22,23].

Most previous studies have applied triclosan directly or after loading into resin materials [24], which showed advantageous physicochemical properties. However, triclosan is a release-type antimicrobial agent and has certain limitations, showing a “burst effect”, concentration dependence, and often depletion during use, with a gradual decrease in the antimicrobial efficiency of the material [25]. Ultimately, these limitations may have a negative impact on the physicochemical properties and mechanical strength of the polymeric material.

It has therefore become particularly important to develop a polymer that reduces the leaching of TCS, maintains a high retention rate, and shows a long-term and sustained antimicrobial effect. In this study, triclosan was modified to make it light-curable and copolymerisable with adhesive, so that the modified antimicrobial monomer TCS-IH would not precipitate over time, thus achieving long-term antimicrobial effect. For the first time, TCS was added to a universal adhesive, and its antimicrobial and mechanical properties were investigated. A pairwise comparison between groups of adhesives with different mass percentages of TCS-IH was also carried out to screen the optimum amount for addition.

## 2. Materials and Methods

### 2.1. Materials

Single Bond Universal (3M ESPE, St. Paul, MN, USA), All Bond Universal (Bisco, Inc., Schaumburg, IL, USA), and Filtek Z350XT (3 M ESPE, St. Paul, MN, USA) were used. Information on these two common commercial adhesives was given in Table 1. 3-(4,5-dimethylthiazole-2yl)-2,5-diphenyltetrazolium bromide (MTT) was purchased from Sigma Aldrich Corp, (St. Louis, MO, USA).Triclosan(TCS), isophorone diisocyanate (IPDI), dibutyltin dilaurate (DBTDL), 2-hydroxyethyl methacrylate (HEMA), dimethyl sulfoxide (DMSO) and acetone were of analytical grade and were provided by Aladdin (Shanghai, China).

### 2.2. Synthesis of Light Curable Triclosan Derivatives

The synthesis uses the method of solution copolymerization. TCS white powder (0.05 mol, 14.45 g) purchased commercially, IPDI (0.1 mol, 22.25 g) and DBTDL (0.05 mol, 0.15 g) were continuously reacted in a three-necked flask with a condensation system and mechanical stirring device under controlled temperature in a water bath at 68 °C for 3 h. At the end of the reaction, the hydroxyl group of the resulting product was detected using FTIR (Nicolet iS 10, Madison, WI, USA). After the disappearance of the hydroxyl group, HEMA (0.1 mol, 13 g) was added dropwise, and a white transparent liquid product was obtained after 6 h of reaction. The product was again examined by FTIR, and the absence of the isocyanate group proved that the reaction was complete with the C=C bond. The product was purified by petroleum ether precipitation, and the white precipitate obtained was again dissolved in tetrahydrofuran. The process was repeated three times, and the solvent was removed by evacuation to give a light curable triclosan derivative. The TCS-IH products were identified by ^1^H NMR (Bruker AVANCEIII500, Rheinstetten, Germany) and FTIR spectroscopy.

### 2.3. Preparation of Antibacterial Adhesive

Single Bond Universal and All Bond Universal were used as carrier bonding systems. TCS-IH were dissolved in acetone solution at 1:4 by mass, after which they were mixed into two universal adhesives at 0%, 1%, 3%, 5%, and 7% by mass to obtain antimicrobial universal adhesives with different components (0 wt%, 1 wt%, 3 wt%, 5 wt%, and 7 wt%) for the following experiments.

### 2.4. Biofilm Experiments

#### 2.4.1. Preparation of Specimens

The composite resin was filled in a polyethylene mould (inner diameter = 1.0 cm, thickness = 1.0 mm), and to prevent the formation of an oxidation-inhibiting layer, a polyethylene film was used to cover the surface, and multipoint overlapping light curing was performed with LED light curing system (Bluephase N, Shanghai, China); output light intensity approximately 1200 mW/cm^2^ for 20 s. A pipette was used to pick up 15 μL of the synthetic antibacterial adhesives of each group, drop them on the round composite resin sheet, and apply the adhesive evenly with a brush. Two kinds of commercial universal adhesives were used as the control group and were prepared in the same way. All the prepared specimens were soaked in distilled water, stirred for 1 h to remove unpolymerized adhesive monomers, dried at room temperature, and sterilized by ultraviolet light for 2 h for use.

#### 2.4.2. Recovery and Culture of Experimental Strains

In the resuscitation experiment, anaerobic *S. mutans* (UA159) freeze-dried strains were inoculated on the surface of brain heart infusion (BHI) agar plates and incubated at 37 °C and 5% CO_2_ for 48 h. Gram-stained smears were examined by microscopy after the strains were cultured. After the strains were identified as pure cultures by morphology, they were serially passaged twice. A disposable inoculating loop was used to pick a single colony, inoculate it in 10 mL BHI liquid medium, and incubate it at 37 °C and 5% CO_2_ for 24 h. After the cultured bacterial suspension reached the logarithmic growth phase, the concentration of the bacterial solution was adjusted to 1 × 10^6^ colony forming units (CFU)/mL using a spectrophotometer before the experiment for use.

#### 2.4.3. Biofilm Formation

The above sterilized specimens were placed in each well of a 24-well plate with the adhesive side facing upwards, and 2 mL of the above prepared bacterial suspension was injected. The specimens and suspension were incubated at 37 °C and 5% CO_2_ for 24 h and then carefully removed and transferred to a new 24-well plate, and each well was slowly reinjected with 2 mL of BHI liquid medium and incubated for 24 h. Finally, the biofilm cultured for 48 h was obtained for use. The biofilm was kept facing upwards throughout the process.

#### 2.4.4. Biofilm Colony Count

The determination of colony forming units was carried out as follows. The surface of each specimen was gently rinsed with sterile phosphate-buffered saline (PBS) (Solarbio, Beijing, China)to remove any loose, unadhered bacteria. The biofilm was collected using mechanical scraping and shaken with a vortex mixer. The biofilm suspension, diluted 1000 times using a stepwise dilution method, was aspirated from 50 μL of biofilm suspension, added dropwise to a BHI agar plate, evenly coated with a disposable coating rod, and incubated at 37 °C and 5% CO_2_ for 48 h. After this period of time, the colonies (CFU) were counted, the count was multiplied by the corresponding dilution multiple. The antibacterial rate was calculated according to the colony count results to express the antibacterial effect of the sample. The formula is as follows:(1)$$R\left(\%\right)=\frac{B-A}{A}\times 100$$

R: antibacterial rate (%); A: the average number of colonies recovered from the 1 wt%, 3 wt%, 5 wt%, and 7 wt% specimens in the experimental group (CFU); B: average number of colonies recovered from specimens in the control group (CFU).

#### 2.4.5. Biomass Biofilm Assays

The specimens were carefully transferred into a new 24-well plate to ensure that the biofilm was facing upwards. The pipette slowly injected 1 mL of 2.5% glutaraldehyde for fixation for 30 min. After absorbing and discarding the fixation solution, the test piece was placed into the fume hood and dried at room temperature for 20 min. The surface of the test piece was slowly washed with PBS (three times), and 1 mL of 0.2% crystal violet solution was added to each hole. The samples were placed in a shaker at 60 r/min for 20 min. The PBS was rinsed three times again to remove the excess dye. Before eluting and quantifying the crystal violet dye related to the biofilm, photos of the specimens containing the biofilm stained with crystal violet were taken (Figure 3A). Then, ethanol was added to each well and incubated for 30 min to dissolve the crystal violet dye. A total of 200 μL was added to the 96-well plate. The absorbance was measured at OD_600 nm_ by a microplate reader.

#### 2.4.6. Biofilm Metabolic Viability

The specimens were carefully transferred to a new 24-well plate, ensuring that the biofilm was facing upwards. One mL of MTT solution was injected and cultured at 37 °C and 5% CO_2_ for 2 h. The specimens were carefully transferred to a new 24-well plate, and 1 mL of DMSO solution was added to dissolve the formed nauseas. The plate was shaken for 15 min, and 200 μL of DMSO solution was removed from each well and transferred to a 96-well plate. The absorbance was measured at OD_540 nm_.

#### 2.4.7. Growth Curves

The specimens were placed into 2 mL of new BHI liquid medium and incubated at 37 °C with 5% CO_2_ for 4 h, 8 h, 12 h, 16 h and 24 h. The biofilm was collected using the mechanical scraping method and shaken by a vortex mixer. Then, 200 μL of bacterial suspension was sucked into a 96-well plate, and the absorbance was measured at OD_600 nm_ using a microplate reader.

#### 2.4.8. Insoluble Extracellular Polysaccharide

The biofilm was collected by mechanical scraping and put into a centrifugal tube containing sterile distilled water. The tube was centrifuged at 4 °C and 12,000× *g* for 5 min. After absorbing and discarding the supernatant, 0.5 mol/L NaOH was slowly added to the precipitate with a pipette gun after centrifugation and centrifuged for 5 min under the same conditions. The supernatant was absorbed and added to 48-well plates with sterile distilled water, 6% phenol (prepared and used now) and concentrated sulfuric acid, and 160 µL was added to each well and cultured for 30 min. The absorbance was measured at OD_490 nm_ by a microplate reader to quantify the amount of biofilm.

#### 2.4.9. SEM

The test piece was carefully transferred to a new 24-well plate to ensure that the biofilm faced upwards. Then, the specimen was immersed in 2.5% glutaraldehyde and fixed for 12 h (at 4 °C). The 30%, 50%, 70%, 80%, 90% and 100% anhydrous ethanol solution gradient was dehydrated with anhydrous ethanol, and each gradient was dehydrated for 10 min. Afterwards, the dry specimen was sprayed with gold and viewed under SEM (SEM; Carl Zeiss, Germany) at 10 kV voltages.

#### 2.4.10. Live/Dead Staining of Bacteria

The specimens were transferred carefully to a new 24-well plate, ensuring that the biofilm was facing upwards. Calcein-AM and propidium iodide (PI) (Solarbio, Beijing, China) were mixed in a 1:1 ratio, and 1 mL of staining solution was slowly added to the wall of the plate using a pipette in a dark room and cultured for 15 min at room temperature. Calcein-AM is a cell staining reagent for fluorescent labelling of living cells, which fluoresces green only through disordered areas of dead cell membranes to the nucleus, where it is embedded in the double helix of the cell’s DNA to produce red fluorescence. Biofilm images were collected using a laser confocal scanning microscope CLSM (Olympus FV1000, Tokyo, Japan). The 3D superimposed maps were drawn using Imaris 9.0 software.

### 2.5. Cytotoxicity Assay

After the above ultraviolet disinfection and sterilization test pieces, the specimens were immersed in cell high glucose medium (1% double antibody, 10% foetal bovine serum buffer), 37 °C, 5% CO_2_ in a cell culture incubator for 24 h, using a suction filter to remove bacteria, and stored in a 4 °C freezer for further use. For the resuscitation of experimental mouse fibroblasts (L929) (Shanghai, China), cells in logarithmic growth phase were inoculated into 96-well plates at a density of 5 × 10^3^ cells per well, and 200 μL of cell high sugar medium was added to each well and incubated for 24 h in an incubator. The cells were observed to grow again from the original wall under the microscope, and the bottom of the well plate was spread by a monolayer of cells. The medium was aspirated and discarded, and the prepared extract was slowly added and coincubated in the incubator for 24 h. The extract was aspirated and discarded, and the surface of the specimen was slowly rinsed with PBS. Ten μL of CCK-8 reagent was added to each well, and its absorbance was measured at OD_450 nm_ using an enzyme standardization instrument. The cellular activity was calculated as follows.
(2)$$\mathrm{Cell}\mathrm{viability}\%=\frac{{\mathrm{OD}}_{\mathrm{Experimental}\mathrm{group}}-{\mathrm{OD}}_{\mathrm{Black}\mathrm{group}}}{{\mathrm{OD}}_{\mathrm{Control}\mathrm{group}}-{\mathrm{OD}}_{\mathrm{Black}\mathrm{group}}}\times 100\%$$

### 2.6. Degree of Conversion

FTIR was used to assess the degree of conversion of each group of specimens. Each group of antibacterial adhesives was coated on the surface of a potassium bromide sheet, and the infrared spectrum before curing was collected and recorded. Each group of five samples was tested. The entire operation was carried out in a dark room to avoid visible light from affecting the adhesive polymerization. The ratio of the absorbance of the aliphatic “C=C” bond absorption peak (1637 cm^−1^) to the absorbance of the aromatic “C=C” bond reference peak (1608 cm^−1^) before and after photocuring was recorded. The ratio is calculated as follows:(3)$$\mathrm{Degree}\mathrm{of}\mathrm{conversion}\left(\%\right)=[1-\frac{(A1637/A1608)\mathrm{cured}}{(A1637/A1608)\mathrm{uncured}}]\times 100\%$$

### 2.7. Microtensile Bond Strength Testing

A total of intact fresh decay-free third molars in the age group of 18–25 years was collected. The prepared dentine planes were all carefully observed under a light microscope to ensure that no residual enamel or pulp cavity was exposed.

The teeth collected were divided into 5 large groups according to the mass percentage of antibacterial filler added. Each large group was further divided into two subgroups according to the different bonding modes, etch-and-rinse bonding mode and self-etch bonding mode. The composition of the adhesive, the method of use, etc., are listed in Table 1. All treatments were performed according to the manufacturer’s instructions. Finally, the composite resin Z350 was built up on the occlusal dentin surface of the specimens using a resin composite which was applied in two layers of 2 mm each and light cured for 40 s.

The subgroups with the two bonding patterns were in turn divided into two subgroups (*n* = 4) according to the testing conditions of whether they underwent thermal cycles. In the thermal cycle group, the prepared teeth were incubated at 5–55 °C for 5000 cycles with a 30 s dwell time in each water bath. The other set was tested with the prepared teeth placed in deionized water stored at 37 °C for 24 h. Then, their microtensile bond strength was tested separately.

The prepared tooth was cut along the long axis of the tooth perpendicular to the bonding interface under the action of a slow-speed cutter. Resin-dentin strips with a cross-sectional area of approximately 1 mm × 1 mm were prepared and used for microtensile bond strength testing.

All the prepared resin dentin strips were glued onto the microtensile testing moulds with cyanoacrylate glue, and tension was applied using a universal testing machine (AG-X plus) at a tensile speed of 1 mm/min until the resin dentin strips fractured. For fractured specimens, the microtensile bond strength was calculated from the cross-sectional area of each resin dentin strip measured individually and in MPa using an electronic micrometer. The microtensile bond strength was calculated as follows:(4)$$\mathrm{Microtensile}\mathrm{bond}\mathrm{strength}\left(\mathrm{Mpa}\right)=\frac{F_{\max}\left(N\right)}{S({\mathrm{mm}}^{2})}$$

### 2.8. Statistical Analysis

The above experimental data were described using the mean ± standard deviation (SD) when they conformed to a normal distribution. The results of antimicrobial experiments (biofilm colony count, biofilm metabolic activity, amount of biofilm formed and insoluble exopolysaccharide), as well as the cytotoxicity and double bond conversion of the experimental adhesive in each group, were analysed by one-way analysis of variance (ANOVA). The effects of different antibiofouling mass percentages (0 wt%, 1 wt%, 3 wt%, 5 wt%, 7 wt%), bonding modes (etch-and-rinse bonding mode and self-etch bonding mode) and testing conditions (24 h and 5000 thermal cycles) on the microtensile bond strength were analysed by three-way ANOVA with Tukey’s test to compare the two sample means between groups, and the interaction between each factor was analysed. Statistical analysis was performed using SPSS statistics, and significance levels were determined as α = 0.05.

## 3. Results

### 3.1. Characterization of TCS-IH

The chemical structure of the obtained TCS-IH was confirmed by ^1^H NMR spectroscopy and FTIR. As shown in Figure 1A, the peaks centred at 5.64 ppm and 6.18 ppm (peak 1) are attributed to the methylene protons at the end of the double bond. The peak centred at 1.45 ppm (peak 2) is attributed to the methylene protons on the six-membered ring. The peaks at 6.85–7.55 ppm (peaks 3–8) were attributed to the methylene protons on the benzene ring of triclosan. The absorption peak at 3370 cm^−1^ in Figure 1B is the characteristic N-H stretching vibration peak in the product from the reaction of isocyanate groups and hydroxyl groups; at 1630 cm^−1^ is the absorption peak of C=C.

### 3.2. Biofilm Colony Count

Figure 2 is a typical picture of bacterial colony growth in biofilms obtained after five groups of two universal adhesive specimens containing different TCS-IHs were cocultured with *S. mutans* for 48 h. Compared with the control group, the number of colonies in all experimental groups decreased significantly, except that there was no significant difference between the 5 wt% and 7 wt% groups (*p* > 0.05), and there were significant differences between the other groups (*p* < 0.05).

Table 2 and Table 3 show the results of the antibacterial rates (%) of each group. The antibacterial rates of the 5 wt% and 7 wt% groups were both more than 99%, which belonged to class I, indicating that these two groups had strong antibacterial properties. The antibacterial rate of the 3 wt% group belongs to class II, which is antibacterial.

### 3.3. Biomass Biofilm Assays

Figure 3A shows the results of quantification of biofilm by crystal violet staining: by measuring the total biofilm amount on the surface of the test pieces of each group of the two universal adhesives after 48 h of coculture, compared with the control group (0 wt%), the biofilm amount of each experimental group (1 wt%, 3 wt%, 5 wt%, 7 wt%) decreased significantly. Although there was no significant difference between the 5 wt% and 7 wt% groups (*p* > 0.05), there were significant differences among the other groups (*p* < 0.05).

### 3.4. Biofilm Metabolic Viability

Figure 3B shows the results of the detection of biofilm metabolic activity by MTT colorimetry: two commercial universal adhesives were used as the control group (0 wt%), and a very high biofilm metabolic activity was detected. With the increase in TCS-IH content, the metabolic activity of biofilms in each experimental group (1 wt%, 3 wt%, 5 wt%, 7 wt%) decreased significantly, and there was a significant difference among the groups (*p* < 0.05).

**Figure 3 polymers-15-00304-f003:**
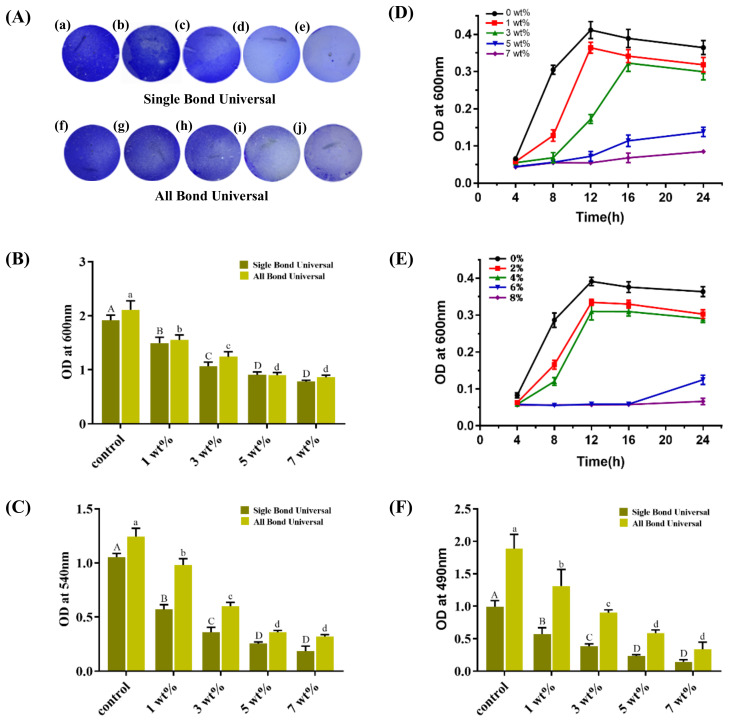
Antibacterial properties of adhesives containing different mass percentages of TCS-IH, (**A**,**B**) are the quantitative detection results of the staining pictures and quantitative test results of production of antibacterial adhesive biofilm in each group by crystal violet staining method. (**a**–**e**) and (**f**–**j**) represent, in turn, the experimental, 1 wt%, 3 wt%, 5 wt%, and 7 wt% groups in the Single Bond Universal and All Bond Universal; (**C**) Results of MTT assay for metabolic activity of biofilm; (**D**,**E**) are the bacterial growth curves after co-culture of Single Bond Universal and All Bond Universal antimicrobial adhesives with *S. mutans*, respectively; (**F**) results of insoluble extracellular polysaccharide assay after co-culture of antibacterial adhesive with *S. mutans* in each group. Data were presented as means ± standard deviations. Different capital letters represent the statistical difference between Single Bond Universal antibacterial adhesives in each group (*p* < 0.05), and different small letters represent the statistical difference between All Bond Universal antibacterial adhesives in each group (*p* < 0.05).

### 3.5. Growth Curves

Figure 3C shows the bacterial growth curve drawn by the average absorbance (OD) values of the biofilm suspension at each measurement time point after the antibacterial test pieces of each group of two universal adhesives were cocultured with *S. mutans* for 48 h. With the increase in the mass percentage of TCS-IH, the OD value of the biofilm suspension at the same time point decreased, and the curve slope of the biofilm suspension in the experimental group (1 wt%, 3 wt%, 5 wt%, 7 wt%) and the time point when it entered the logarithmic phase decreased and lagged behind the time point when it entered the logarithmic phase in the control group (0 wt%). The bacterial density in the biofilm suspension decreased with increasing mass percentage of TCS-IH, and the inhibition of bacterial growth by the addition of TCS-IH was concentration-dependent.

### 3.6. Insoluble Extracellular Polysaccharide

By measuring the absorbance of insoluble extracellular polysaccharides, the biofilm on the surface of each group of test pieces of two universal adhesives was quantified. Figure 3D shows the results. Compared with the control group (0 wt%), the amount of biofilm in each experimental group (1 wt%, 3 wt%, 5 wt%, 7 wt%) decreased significantly, except that there was no significant difference between the 5 wt% and 7 wt% groups (*p* > 0.05), and there were significant differences among the other groups (*p* < 0.05).

### 3.7. SEM

Figure 4 shows the observation results of biofilms on the surface of antibacterial test pieces of two universal adhesives under SEM after coculture with *S. mutans*. With the increase in the mass percentage of TCS-IH, the amount of *S. mutans* bacteria on the surface of the test piece gradually decreased, the 3D structure disappeared, the surface morphology of *S. mutans* was no longer smooth, rough and concave, the integrity of the cell membrane was damaged, and the contents flowed out.

### 3.8. Live/Dead Staining of Bacteria

Figure 5 shows the 3D images of five groups of biofilms with different content of TCS-IH universal adhesives. The control group was stained mainly with green fluorescence, the biofilm on the surface of the test piece was dominated by living bacteria, and the biofilm was thick and dense. In the 1 wt% and 5 wt% groups, the number of dead bacteria on the surface of adhesive specimens containing TCS-IH increased, showing an orange/yellow area with red and green superimposed. With the increased TCS-IH content, the specimen surface was mainly stained with red fluorescence, and the dead bacteria dominated the biofilm. The biofilm became thinner, and the number of bacteria decreased. The results showed that TCS-IH inhibited the formation of biofilms and had a better antibacterial effect.

### 3.9. Cytotoxicity Assay

Figure 6A shows the cytotoxicity results of the five groups containing two universal adhesives with different TCS-IHs after coculture with L929 cells for 24 h through a CCK-8 assay. Compared with the control group, the experimental adhesives in each group showed no significant difference (*p* > 0.05), and the cell viability was above 80%. The adhesives in each group had no obvious cytotoxicity.

### 3.10. Degree of Conversion

Figure 6B shows the results of five groups of degrees of conversion of two universal adhesives containing different TCS-IH. The degree of conversion rate of the adhesive in the experimental group was not affected by the addition of TCS-IH, and there was no significant difference among the groups (*p* > 0.05).

### 3.11. Microtensile Bond Strength

Table 4 and Table 5 show the results of the bond strength of the two universal adhesives, examined by using different bonding modes and test conditions. Mass percentages: 7 wt% were significantly different from the other groups in the self-etch bonding mode, Single Bond Universal 24 h and 5000 thermal cycles, and All Bond Universal 24 h (*p* < 0.05). The other groups were not significantly different (*p* > 0.05). Bonding mode: There was a significant difference between the 24 h Single Bond Universal and All Bond Universal groups at the same mass percentage for both bonding modes (*p* < 0.05). There was a significant difference between the Single Bond Universal 7 wt% group after 5000 thermal cycles (*p* < 0.05). The other groups were not significantly different (*p* > 0.05). Test conditions: After 5000 thermal cycles of the same mass percentage, only the etch-and-rinse bonding mode was significantly different between the two groups (*p* < 0.05), and the self-etch bonding mode was not significantly different between the two test conditions (*p* > 0.05).

## 4. Discussion

To obtain better dentin bonding, different types of new adhesives and bonding strategies have emerged. The existing adhesives have been developed to eighth-generation universal adhesives. Because of its simple operation steps and flexible operation methods, as well as the advantages of “multipurpose, multifunction and multipurpose”, it is now increasingly widely used in clinical practice. The adhesive itself has no inherent antibacterial property and is used in a complex oral environment, such as repeated chewing force, changing temperature, mismatched thermal expansion coefficient between the material and tooth hard tissue, etc., which easily lead to the attachment of plaque biofilm, thus affecting the durability of bonding repair. Studies have also shown that [26], compared with normal dentin, the degradation of the carious dentin mixed layer is more obvious. Therefore, the addition of antibacterial substances in universal adhesives is very important to reduce the incidence of dental caries and improve the stability and durability of the adhesive interface.

TCS is a broad-spectrum antibacterial agent that can act on both gram-positive and gram-negative bacteria. Triclosan exerts its antibacterial action through the following substrates: (1) inhibition of protein reductase in bacterial fatty acid biosynthesis. Fatty acid synthesis is essential in bacteria, and inhibition of its synthesis leads to bacterial cell membrane rupture and bacterial decomposition [27]. (2) Direct action on the cell membrane alters the permeability of the cell membrane, causing leakage of cytoplasm and leading to bacterial death [28]. In previous studies, TCS has been used in composite resins, showing reliable physical and chemical properties, no cytotoxicity, and good biosafety [29]. Therefore, in this study, TCS was added to universal adhesives, Single Bond Universal and All Bond universal to prepare adhesives with antibacterial activity. However, TCS is a releasing antibacterial agent that has a “burst effect”. If it is directly mixed with the universal adhesive, TCS will gradually leach out over time, which will adversely affect the long-term antibacterial activity or mechanical properties of the universal adhesive. To eliminate this defect, we modified TCS and introduced a C=C bond to endow it with light curable properties. After light curing, the modified TCS can be successfully included through covalent linkages inside the universal adhesive to form a dense grid structure, in order to obtain long-term antibacterial effect, good mechanical properties and biological safety [30]. In addition, some studies have observed that the polymerizable monomers covalently linked inside the adhesive system furnish a better control of the antibacterial activity by preventing the leaching of the antibacterial agent from the polymerized matrix [31].

The modified TCS did not affect its antibacterial activity with the introduction of a light curable C=C bond structure. Muciniphila in each experimental group showed concentration-dependent inhibitory effects on biofilm formation, exopolysaccharide synthesis and bacterial metabolic activity in biofilms. The antibacterial effective group (C-Cl) in TCS was not affected, which was consistent with the findings of Wu et al. [32]. The research of Machado et al. [24] similarly proved that TCS could exert a better antibacterial effect in oral materials.

In this experiment, the experimental methods of colony counting, crystal violet staining and drawing growth curves were used to evaluate the number of bacteria in the biofilm on the surface of each group of specimens and their growth and reproduction. Each individual colony on an agar plate is a reproduction of a single bacterium. Therefore, the number of bacteria on the surface of the test piece can be converted according to the dilution factor and the amount of sampling inoculum. Numerous bacterial colonies were formed in the 0 wt% group. With the increase in the TCS-IH content, the number of colonies decreased significantly, especially the 5 wt% and 7 wt% groups, which had an antibacterial rate of over 99% against *S. mutans*. Crystal violet staining is a simple and quick experimental method to detect bacterial biofilms. Crystal violet is a basic dye that can combine with negatively charged extracellular polymeric substance (EPS) to give a blue-violet colour [33], and can be dissolved from biofilms by 95% alcohol. Finally, the absorbance value was measured to assess the amount of biofilm formation. From the colour gradient of each group of specimens stained with crystal violet in Figure 3A and the absorbance value in Figure 3B, TCS-IH has been shown to inhibit the formation of plaque biofilms, especially the 5 wt% and 7 wt% groups, which showed better antibacterial properties. The growth curve can reflect the growth and reproduction of the bacterial population. Bacteria in the logarithmic phase grow and reproduce quickly and have a strong metabolism. Under the action of TCS-IH, the time and reproduction speed of bacteria in each experimental group were lower than the time and reproduction speed in the control group.

Crystal violet is a nonspecific stain, and both live and dead bacteria in the biofilm can be stained with it. Therefore, to further explore the metabolic activity of TCS-IH on *S. mutans* biofilms, we chose the MTT reduction method. The principle of the MTT method is that succinate dehydrogenase in bacterial cells can reduce light yellow MTT to purple formazan [34], dissolve the formazan crystals in DMSO, and finally determine the value of the measured absorbance. The activity of the dehydrogenase system in bacteria was detected, and the metabolic activity of *S. mutans* biofilms in different groups was quantitatively assessed. Exopolysaccharides, especially glucans, have a great influence on the cariogenicity of *S. mutans*. In *S. mutans* plaque biofilms, water-insoluble glucan was identified as the main component of exopolysaccharides [35]. The gradual decrease in the absorbance values of each group in Figure 3C,F also proves that TCS-IH also inhibits the metabolic activity of active bacteria and the production of extracellular polysaccharides in the biofilm.

In the SEM micrographs, the number of *S. mutans* on the surface of each experimental group was significantly reduced, the clustered network distribution and three-dimensional structure were no longer obvious, and a dispersed short-chain distribution was observed. The bacteria shrank, and the surface was uneven, no longer showing the original full chain structure. Confocal laser scanning microscopy (CLSM) can capture images of biofilms at all levels in real time and perform 3D reconstruction to intuitively understand the number and distribution of live and dead bacteria. Typical images of live/dead staining of *S. mutans* biofilms are shown in Figure 5. The biofilms of the control group were all relatively intact, and only a very small number of bacteria died, and dead bacteria appeared inside the biofilms of the TCS-IH adhesive added to the 1 wt% group and 3 wt% group, showing the colour of the yellow red interphase, indicating that low concentration TCS-IH had a limited ability to disrupt the deep-layer biofilms, whereas with the increase in TCS-IH content, the plaque biofilms all exhibited a large amount of dead red. The TCS-IH antibacterial system was demonstrated to be able to disrupt the formed biofilms of *S. mutans* more effectively. SEM and CLSM images confirmed the remarkable antibacterial effect of TCS-IH. All the above experimental results showed that the adhesive with 5 wt% and 7 wt% mass percentages of TCS-IH exhibited significant antibacterial properties against *S. mutans*.

In addition to secondary caries, bond strength is also an important condition to ensure the durability of bonded restorations. This experiment compares the effect of different mass percentages of TCS-IH, produced in different bonding modes and testing conditions, on the bond strength of a universal adhesive, Single Bond Universal and All Bond Universal, and also evaluating the degree of conversion. The experimental results showed that there was no significant difference in the degree of conversion between the groups (*p* > 0.05).

Three variables were involved in this experiment, namely, the content of TCS-IH (0 wt%, 1 wt%, 3 wt%, 5 wt%, 7 wt%), the bonding mode (etch-and-rinse and self-etch) corrosion and the test conditions (24 h and thermal cycling). The results in Table 4 and Table 5 show that three factors have an impact on the bond strength of Single Bond Universal and All Bond Universal.

Under the test condition of water storage for 24 h, the bonding strength of each group of adhesives in the two bonding modes was significantly different (*p* < 0.05), and the bonding strength in the etch-and-rinse bonding mode was significantly higher than the bonding strength in the self-etch bonding mode, which is consistent with the research results of Hass et al. [36]. on the bonding strength of different bonding modes, probably because the dentin surface is formed with a thickness of approximately ≥5 μm after 37% phosphoric acid etching. In the mixed layer, the depth of demineralization is large, the penetration of the adhesive between the dentin collagen fibres is greater, the formed resin tag is longer, and the bonding strength is better. Self-etch adhesives have limited acid etching and penetration capabilities. The pH of Single Bond Universal is approximately 2.7, and the pH of All Bond Universal is approximately 3.2. The adhesives are ultraweak acid self-etch adhesives, the thickness of the mixed layer formed is relatively thin, the thickness is approximately 0.2–0.5 μm [37], the resin tag is also short, and the bonding strength is lower than the bonding strength of the etch-and-rinse bonding mode.

Among the various mechanisms of hybrid layer degradation documented in the literature [38,39], the hydrophilicity of many single-bottle adhesives has become one of the important factors in their susceptibility to degradation. After polymerization, they may be more likely to absorb moisture, resulting in hydrolytic effects.

After thermal cycling, the bond strength in both bond modes was compromised, but in terms of the magnitude of the decrease, the decrease was greater in the etch-and-rinse bonding mode. The bonding strength of the self-etch bonding mode is less affected, probably because after the etch-and-rinse bonding mode, the dentin collagen fibres in the demineralization area at the bottom of the hybrid layer are not fully infiltrated by the bonding agents, and there is a phenomenon of incoordination of osmotic demineralization, making it a “weak structure”, which is gradually occupied by water and accelerates the precipitation of the bonding agent monomers [40]. Meanwhile, proteolytic enzymes in the dentin matrix were activated to break down the ester bonds of naked collagen fibres and adhesive agents, resulting in a significant decline in dentin bonding strength.

For clinical applications, materials must have good biocompatibility. According to this CCK-8 cytotoxicity experiment, the cell viability of each group was more than 90%, which confirmed that the addition of TCS after modification, without obvious cytotoxicity, had a better biosafety.

In summary, the antibacterial monomer TCS-IH was synthesized in this experiment and added to the Single Bond Universal and All Bond Universal with different mass percentages to obtain adhesives with antibacterial activity.

According to the above experimental results, to maintain the dentin bond strength with long-term antibacterial activity, 5 wt% antibacterial monomer TCS-IH was selected as the addition in the adhesives.

## 5. Conclusions

The present study reported the synthesized 5 wt% light curable triclosan derivatives that were incorporated into a universal adhesive with favourable antimicrobial capabilities to combat plaque biofilms. At the same time, it showed reliable adhesive properties, which facilitated the stabilization of the adhesive-dentin bonding interface and could significantly improve the service life of oral restorations.

## Figures and Tables

**Figure 1 polymers-15-00304-f001:**
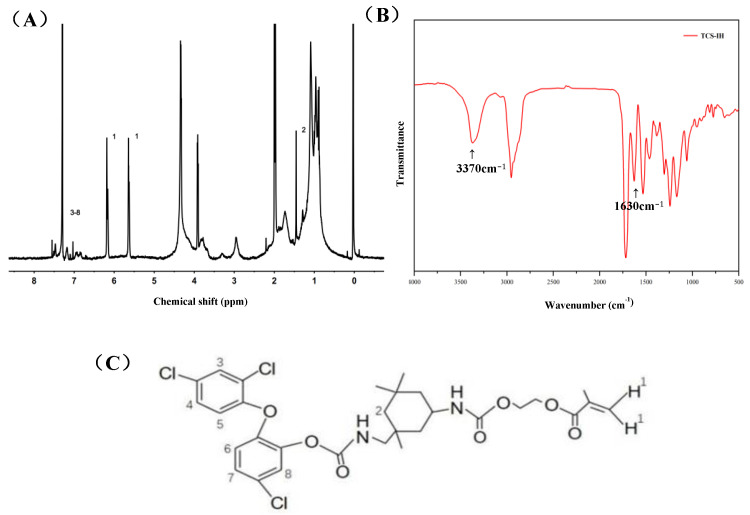
1H NMR (**A**) and FTIR (**B**) of TCS-IH (**C**).

**Figure 2 polymers-15-00304-f002:**
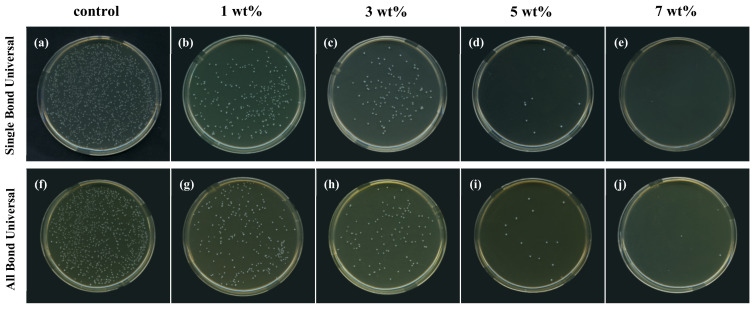
Colony counts images of *S. mutans* after co-culture with SingleBond Universal (**a**–**e**) and All Bond Universal (**f**–**j**) groups of antimicrobial bonding agents.

**Figure 4 polymers-15-00304-f004:**
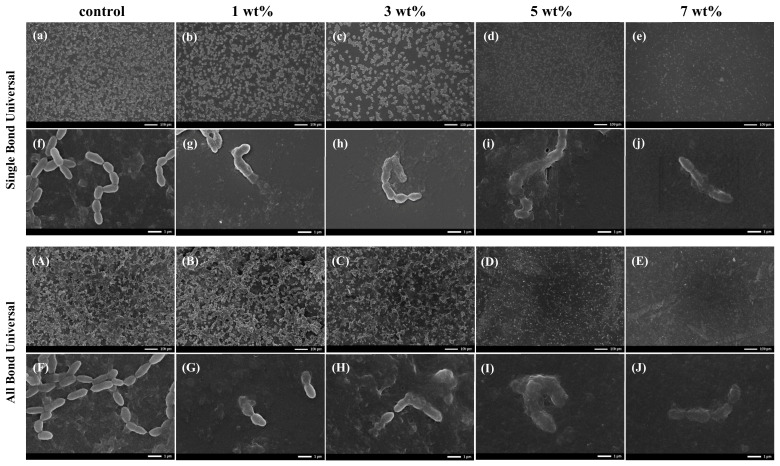
SEM images of bacterial morphology on the surface of specimens co-cultured with antibacterial adhesive and *S. mutans* in Single Bond Universal and All Bond Universal. In figure (**a**–**e**) and (**A**–**E**), with the increase in antibacterial TCS IH content, the number of bacteria on the surface of the test piece gradually decreases, and the aggregation gradually becomes dispersed (10 kV; ×100). In Figure (**f**–**j**) and (**F**–**J**), the *S. mutans* cell membrane in the control group is smooth and complete. With the increase in TCS IH content, the *S. mutans* cell membrane gradually shrinks, becomes rough and uneven, the cell membrane is broken, and the content flows out (10 kV; ×10,000).

**Figure 5 polymers-15-00304-f005:**
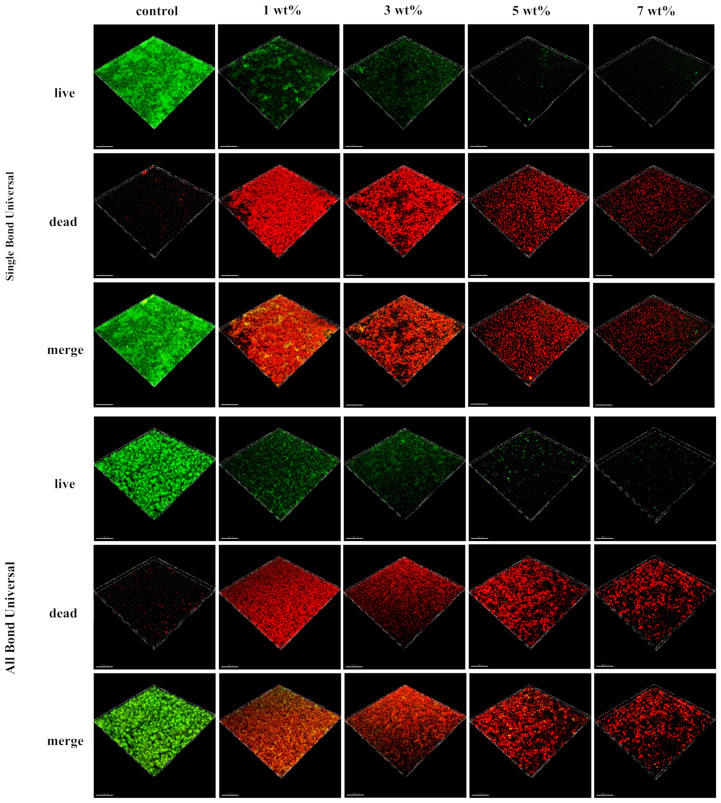
Representative three-dimensional live/dead images of biofilm after co-culture of antibacterial adhesive and *S. mutans* in Single Bond Universal and All Bond Universal. The picture from top to bottom is: live bacteria stained with green fluorescence; Dead bacteria stained with red fluorescence; The combined graph of the two overlays.

**Figure 6 polymers-15-00304-f006:**
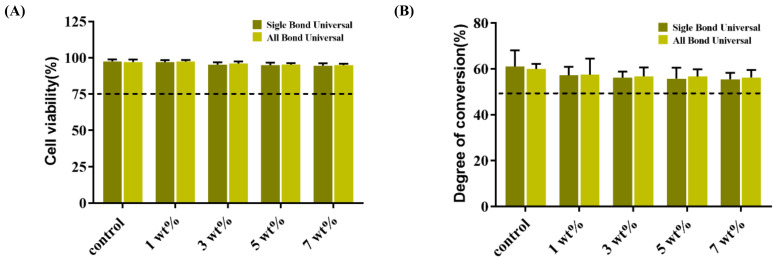
(**A**,**B**) are the results of the cytotoxicity and degree of conversion rate of each group of antimicrobial adhesive are shown respectively.

**Table 1 polymers-15-00304-t001:** Adhesive system, composition and application mode of the adhesive systems used according to the manufacturer’s instructions.

Adhesive	pH	Composition	Manufacturer	Self-Etch	Etch-and-Rinse
Single Bond Universal	2.7	bis-GMA, HE-MA, 10-MDP, di-methacrylate, ethanol, water, resins, silane, filler, initiators, Vitrebond copolymer	3 M ESPE, USA	Apply the adhesive or adhesive mixture to the prepared tooth and rub it in for 20 s. Gently air dry the adhesive for approximately 5 s for the solvent to evaporate. Light cure for 10 s.	Apply etchant for 15 s. Rinse thoroughly with water and dry with water-free and oil-free air or with cotton pellets; do not over dry. Apply adhesive as for the self-etch mode
All Bond Universal	3.2	Bis-GMA, 10-MDP, HEMA, ethanol, initiators, water	Bisco Inc., USA	Apply two separate coats of adhesive with agitation for 10–15 s per coat. Evaporate solvent by thorough airdrying using an air syringe for at least 10 s. There should be no visible movement of the adhesive. The surface should have a uniform glossy appearance. If not, repeat steps 1 and 2. Light cure for 10 s.	Etch for 15 s. Rinse thoroughly. Remove excess water by blotting the surface with an absorbent pellet or high volume evacuation for 1–2 s, leaving the preparation visibly moist Apply adhesive as for the self-etch mode.

**Table 2 polymers-15-00304-t002:** Colony counts and antibacterial rate (%) of *S. mutans* after co-culture with Single Bond Universal containing different mass ratios of TCS-IH. ($\bar{x}$ ± s, *n* = 5).

Group	Control	1 wt%	3 wt%	5 wt%	7 wt%
Colonies	1378 ± 45.4 ^a^	233 ± 18.7 ^b^	109 ± 14.5 ^c^	10 ± 3.2 ^d^	2 ± 0.8 ^d^
Antibacterial rate	0.0	83.1	92.1	99.3	99.9

Different lowercase letters (comparing different mass ratios of TCS-IH) indicate statistical differences (*p* < 0.05).

**Table 3 polymers-15-00304-t003:** Colony counts and antibacterial rate (%) of *S. mutans* after co-culture with All Bond Universal containing different mass ratios of TCS-IH. ($\bar{x}$ ± s, *n* = 5).

Group	Control	1 wt%	3 wt%	5 wt%	7 wt%
Colonies	1107 ± 51.3 ^a^	259 ± 21.3 ^b^	107 ± 31.5 ^c^	10 ± 2.1 ^d^	2 ± 1.1 ^d^
Antibacterial rate	0.0	76.6	90.3	99.1	99.8

Different lowercase letters (comparing different mass ratios of TCS-IH indicate statistical differences (*p* < 0.05).

**Table 4 polymers-15-00304-t004:** Microtensile bond strength of Single Bond Universal in different bonding strategy and test conditions (MPa, $\bar{x}$ ± s, *n* = 16).

Test Conditions	Mass Ratio	Etch-and-Rinse	Self-Etch
24 H	0 wt%	40.3 ± 9.7 ^Aa^*	34.4 ± 9.6 ^Ab^
	1 wt%	39.7 ± 13.3 ^Aa^*	33.7 ± 6.1 ^Ab^
	3 wt%	38.3 ± 9.4 ^Aa^*	32.4 ± 7.0 ^Ab^
	5 wt%	37.4 ± 10.6 ^Aa^*	31.5 ± 10.1 ^Ab^
	7 wt%	36.2 ± 13.3 ^Aa^*	29.1 ± 4.4 ^Bb^
5000 Thermal Cycles	0 wt%	34.2 ± 8.8 ^Aa^	30.1 ± 4.5 ^Aa^
	1 wt%	33.1 ± 6.2 ^Aa^	28.6 ± 5.4 ^Aa^
	3 wt%	31.8 ± 7.9 ^Aa^	27.4 ± 3.8 ^Aa^
	5 wt%	30.7 ± 10.0 ^Aa^	26.0 ± 3.0 ^Aa^
	7 wt%	29.7 ± 8.6 ^Aa^	23.8 ± 5.3 ^Bb^

Different capital letters in the same column indicate statistical differences (*p* < 0.05); Different lowercase letters in the same row indicate statistical differences (*p* < 0.05); * in the same column indicates a statistical difference between groups of the same mass ratios (*p* < 0.05).

**Table 5 polymers-15-00304-t005:** Microtensile bond strength of All Bond Universal in different bonding strategy and test conditions (MPa, $\bar{x}$ ± s, *n* = 16).

Test Conditions	Mass Ratio	Etch-and-Rinse	Self-Etch
24 H	0 wt%	44.9 ± 12.1 ^Aa^*	40.4 ± 10.6 ^Ab^
	1 wt%	43.4 ± 9.9 ^Aa^*	39.7 ± 10.7 ^Ab^
	3 wt%	41.8 ± 10.1 ^Aa^*	37.3 ± 8.7 ^Ab^
	5 wt%	40.6 ± 9.1 ^Aa^*	35.9 ± 10.0 ^Ab^
	7 wt%	39.8 ± 11.6 ^Aa^*	34.6 ± 8.9 ^Ab^
5000 Thermal Cycles	0 wt%	37.6 ± 10.5 ^Aa^	35.4 ± 9.2 ^Aa^
	1 wt%	36.2 ± 11.7 ^Aa^	33.7 ± 11.0 ^Aa^
	3 wt%	34.2 ± 8.3 ^Aa^	32.0 ± 10.3 ^Aa^
	5 wt%	33.2 ± 9.7 ^Aa^	31.0 ± 9.1 ^Aa^
	7 wt%	31.3 ± 8.7 ^Aa^	28.1 ± 6.4 ^Ba^

Different capital letters in the same column indicate statistical differences (*p* < 0.05); Different lowercase letters in the same row indicate statistical differences (*p* < 0.05); * in the same column indicates a statistical difference between groups of the same mass ratios (*p* < 0.05).

## Data Availability

The data will be shared on reasonable request to the corresponding author.

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
