# Peer review of "Triclosan to Improve the Antimicrobial Performance of Universal Adhesives"

_polymers, 2023, doi:10.3390/polym15020304_

Round 1

Reviewer 1 Report

1) Page 3, materials and methods part: Here, Materials part is missing. Please add the materials one by one and with the purchasing company information. for example HEMA or single bond universal; where did you purchase them needs to be written.

2) Same page, same part: Please add "mmol or mol and weight percentage'' calculations to the experimental procedure part. So far, only the weight of the products are added but the molar ratios and weight percentages are missing.

3) page 3, preparation of adhesives; do you use any solvent there? If yes, please specify.

4) Page 3: preparation of specimens; you wrote there "light curing" can you please give more details, which light? UV? and also more details about the equipment, brand? the values of light intensities? What is the mechanism of this curing?

5) same part as above: you perform a light curing and then you sterilize the specimens with ultraviolet light for 2 hours. Here more info needs to be given: The sterilization does not interfere with the photocuring? 

6) Which FE-SEM you are using? Please specify the brand and give info about the working voltage? More details about SEM device is needed, such as are you working on backscattering mode?

7) In this paper you are adding Triclosan derivatives to the carrier bonding systems as you mentioned. However, the explanation about the reaction between the Triclosan derivative and carrier bonding material is missing. Can you please explain. Moreover, it needs to be also proved that the triclosan is incorporated successfully into the carrier bonding. Do you show the result of this incorporation in this paper? As I understood, in Figure 3 you are showing the Triclosan characterization, but it is still missing there the control sample; can you please some proof relating before and after triclosan incorporation?

8) It is important here to discuss if the Triclosan is leaching or not from the system? How do you kill the bacteria? With leaching? More explanations and experiments need to be added for this reason. How are you sure the Triclosan incorporated to the carrier is killing the bacteria but not free-Triclosan left in the system?

Author Response

请参阅附件。

Reviewer 2 Report

abstract section : results section can include about % weight of 5% before concluding it in conclusion section.

materials and method section: remove name of the institution

                                                  specify light curing unit used

                                                   what does composted teeth mean

                                                   remove term in vivo for 'the prepared teeth

                                                   in vivo'

Conclusion section: please write it according to the findings/results.

Reviewer 3 Report

The manuscript is very well written and clear. The subject is very relevant and current. In the Introduction the authors might consider removing the paragraph 2, which describes the composition of universal adhesive systems (presence of 10-MDP and PAC).  The methodologies are adequate and results are clear. The n used for the microtensile bond strength test was very small (n=4). I suggest presenting the calculation of the power of the sample. In the results, table 3 did not present footnote to explain the letters (equal or different). Important topics were discussed and the conclusions are adequate.

Round 2

Reviewer 1 Report

Dear MDPI,
I am not satisfied with the following explanations from authors, still missing some important points.

Regarding question 7;
7a) Can you please highlight where is the control group here in NMR or in FTIR? You only have the characterizations of Triclosan added system, I cannot see the control. 

7b) The explanation about the reaction between the Triclosan derivative and carrier bonding material is missing. Can you please explain. It is a covalent bonding? Crosslinking? It is Michel addition? It is which kind of addition? The paper needs more detailed explanation from chemistry/reaction point of view, and most importantly citations are missing. This explanation and citations also need to be added in the manuscript itself.

Question 8) Did you measure the leaching? How can you be sure? Do you have any proof of measurements? That point needs to be clarified by additional experiments. What is a burst affect? Citations are missing definitely

Question 6) Voltage values also need to be added in the methods section.

Author Response

please see details from the attachment
